# Economic and Financial Feasibility of a Biorefinery for Conversion of Brewers' Spent Grain into a Special Flour

Iliane Colpo [1,*], Denis Rasquin Rabenschlag [1], Maria Soares de Lima [2], Mario Eduardo Santos Martins [3] and Miguel Afonso Sellitto [2]

1  Department of Production Engineering, Federal University of Santa Maria, Santa Maria 97105-900, Brazil; drr.ufsm@gmail.com
2  PPGEPS, UNISINOS, São Leopoldo 93022-750, Brazil; etica.consultoriaempresarial.m@gmail.com (M.S.d.L.); sellitto@unisinos.br (M.A.S.)
3  Department of Mechanical Engineering, Federal University of Santa Maria, Santa Maria 97105-900, Brazil; mario@mecanica.ufsm.br
*  Correspondence: ilicolpo@gmail.com

**Abstract:** This study aims to evaluate the financial and economic feasibility of implementing a biorefinery to process the solid waste, called brewers' spent grain, generated in the production of craft beer into special flour. In addition, to present a path for open innovation in the possibility of replication of the process and technology used in the plant. The inappropriate disposal generates an environmental problem, but individually, depending on the production volume of the brewery, the cost of processing the waste can be unfeasible. On the other hand, such waste embeds a high nutritional value for human food. This study followed the precepts of the circular bio-economy and industrial symbiosis strengthening of sustainable development. The research method is the Monte Carlo simulation, including four different scenarios and projections. The results indicate the financial and economic viability of industrial plants—biorefineries—for the transformation of the residue into special flour in three of the four scenarios studied in the five-year cycle. In the Monte Carlo simulation, no losses are evident in any of the 10,000 interactions. The sensitivity analysis demonstrates that the sensitivity of the supply is slightly higher than the price of the final product. Results may be useful to support the development of new, innovative products relying on collaboration among internal and external partners and open innovation concerns.

**Keywords:** economic feasibility; open innovation; brewers' spent grain; special flour; biorefinery

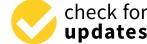



## 1. Introduction

Industries are important social segments and major generators of waste. They are subjected to pressure from organizations, governments, and consumers regarding environmental liabilities [1]. Such pressure reflects on the sustainability performance [2–4] as well as the market share, as many customers prefer buying from environmentally friendly companies [5–7]. Specific market segments also prefer differentiated products that are more nutritious and healthier [8,9] or produced by small companies whose business strategy relies on intrinsic expertise [10].

However, progress toward sustainable development has been slow, requiring more strategic guidance [11]. In developing countries, the effect of sustainable practices is slower than that observed in developed markets [12] due to a lack of resources to implement green practices [13]. Brazilian small and micro enterprises (SME), despite their economic and social potential [14,15], lack support to face environmental challenges [16,17]. SMEs represent more than 90% of the industries in Brazil [15]. Although difficulties may arise in matching social and environmental concerns [18], the reduce-reuse-recycle approach seems to be increasingly adopted by a significant number of organizations [19]. Companies

understand that social and environmental problems require changing the strategy of organizations and introducing interdisciplinary actions and methods [20].

In line with this proposition, several studies highlight that companies may benefit from sustainable solutions. The resolution of problems may involve opportunities to find new customers, innovate the value proposition, collaborate with partners, and develop new, more sustainable, and collaborative business models [21–27].

Brazilian craft beer manufacturers, mostly SMEs, offer a differentiated product with an ever-growing production volume [28]. The main solid byproduct generated from the activity is brewers' spent grain (BSG), which contains a great nutritional potential for human food [29–31]. The BSG contains about 85% moisture, and although a viable and economical raw material for several bio-based and bioenergy products, it has a low value from the industrial perspective [32].

Producers in the state of Rio Grande do Sul, Brazil consider the cost of the waste conversion to be unviable. Such companies usually route the waste to rural producers or dump it in landfills [33]. This waste can be processed and transformed into various products, among which spent grain flour stands out. The transformation process is simple, and the flour presents an interesting option for the bakery industry or even consumers increasingly concerned with the nutritional value of products [8].

In this context, the implementation of a biorefinery, a platform for processing biomass, can solve the waste management problem [34]. This is also valid for a cluster of SME artisanal breweries in Porto Alegre, Brazil. Biorefineries employ various conversion technologies to process waste and by-products [35,36]. It is also possible to employ a circular economy structure in biowaste biorefineries as a sustainable approach towards the circular economy [37].

In Brazil, the National Solid Waste Policy (NSWP) supports circular economy models, establishing a shared responsibility for the preservation of the environment and making manufacturers responsible for the life cycle of their products. It also establishes the following waste management priorities: non-generation, reduction, reuse, recycling, solid waste treatment, and adequate waste disposal [38]. However, small volumes of non-hazardous industrial waste can be disposed of as common waste [39], which is the case in many craft breweries, whose waste volume is small.

Stimulating the use of waste from a business cluster of craft brewers and transforming them into raw material for other production chains can be considered a precursor of industrial symbiosis (IS). IS is related to strengthening the circular economy [40–42].

This paper presents a model for opening up the internal data of the biorefinery plant and the conversion technology used in the process. Open innovation activities can effectively deal with resource and environmental externalities and then relatively balance the economic value and green value of organizations, which is an effective green governance mode [43].

The goal of open innovation is to capitalize on the discoveries and innovations of others in the innovation process, as opposed to closed processes in which companies operate solely on their ideas, capabilities, and professional skills [44].

Open innovation as a paradigm assumes that companies can and should use external and internal ideas, as well as internal and external paths to the market [45,46]. The inclusion of civil society is also emphasized in the development of alternative partnerships and user innovation. Challenges that vary and depend on site circumstances accompany the implementation of collaboration and open innovation approaches [47].

The SMEs play an increasingly important role in the world economy and technological innovation [48], whereas the lessons learned from large companies cannot be easily transferred to SMEs [49,50]. Therefore, the adoption of open innovation in SMEs requires further exploration since SMEs can perform open innovation for the whole process of creation and operation and thus ensure market success [51]. However, biorefineries expect returns on investment and the creation of value for stakeholders, which gives rise to the following research question for this study: Is a spent grain biorefinery in the Porto Alegre-RS craft

breweries cluster economically and financially feasible? The purpose of this study is to evaluate the financial and economic feasibility of a biorefinery to process BSG from the production of craft beer into special flour. In addition, to present a path for open innovation in the possibility of replication of the process and technology used in the plant. To serve as a stimulus for investors interested in sustainable business and entrepreneurs of small breweries that have BSG as their main waste without recognizing its value and the environmental damage it can cause if disposed of inappropriately. BSG has been widely explored in the literature [32]. The authors point out that Brazil is a productive country in terms of publications on this subject, although few studies contemplate biorefineries or industrial applications. Most studies focus on techniques for higher yield and viability of BSG transformation into bioethanol [52–54], biogas [55], and further protein extraction [56–60]. One of the implications of the study is to support entrepreneurship in the craft beer chain, mainly concerning BSG. Craft breweries do not generate large amounts of waste as do large traditional breweries, but with the growing number of companies, a feasibility study may represent a more sustainable way of managing their waste. Biorefineries can become an important strategy for minimizing environmental, social, and economic problems in emerging markets and contribute to preventing irregular waste dumping.

## 2. Theoretical Framework

### 2.1. Sustainable Development and the Circular Economy Model

Sustainability and sustainable development are recurrent themes in the literature. Despite the lack of consensus on the terms due to different interpretations and associations depending on the scenario and areas of activity [61,62], general acceptance is in line with the search for harmony between the needs of humans and the environment [63]. Sustainability is a political vision of society with a focus on preventing the depletion of natural resources. Sustainable development is a collective process of society involving various parties with different powers and interests. In short, sustainable development is a way to achieve sustainability [64,65].

Today, sustainable development is a central concept within the global development policy and agenda, which seeks a mechanism of interaction between society and the environment with the claim of not offering risks or damages to the future and providing for the improvement of the quality of life [66]. The United Nations (UN) has projected the definition of sustainable development globally as development that meets the needs of the present without compromising the ability of future generations to meet their own needs [67].

Sustainable development must be economically efficient, socially inclusive, and ecologically correct, supported by an integrated management system [68]. For example, in the study of a cash transfer program in Mexico for poverty reduction, Alix-Garcia et al. showed that growth in household income caused an increase in the ecological footprint due to poor access to sustainable markets [69].

Businesses play an important role in the pursuit of sustainable development, recognizing their importance in the 2030 Agenda, and are called upon to play their part in achieving the seventeen Sustainable Development Goals, with special recognition for their potential in inclusive economic growth, job creation, and productivity [70]. The circular economy model facilitates the achievement of sustainable development [71].

The concept of circular economy is linked to the optimization of resources [72] that requires maximizing the use of waste as inputs for other processes [73,74]. Additionally, increasing the efficiency of using virgin materials may also contribute to a sustainable society [75] by reducing the need for raw materials [76,77].

The concept includes economic, environmental, and social areas while collecting ideas from various fields that include industrial symbiosis (IS) [78,79], cleaner production, ecology industrial, urban metabolism, biomimicry, and design [80]. IS seeks synergy in firm networks that can foster eco-innovation [81] and long-term cultural change [82] by transforming the current linear economic-based production system toward increased input

circulation and decreased natural resource sequestration. Thus, IS strengthens the CE model by turning physical resources into economic benefits [40,41]. CE includes all the following three sustainability dimensions: economic, environmental, and social areas, while bringing together ideas from various fields, such as IS [42].

IS, conceptually, has more emphasis on the sharing or exchanging of physical resources, such as materials, water, and energy. Other resources could also be traded or shared, such as knowledge, customer relationships, physical structures, workforce, logistics, or agreements between firms that generate resource efficiency and should also be considered as IS [83–85]. Chertow adopted a 3-2 heuristic as a starting point, where at least three different entities must be involved in the exchange of at least two different resources, with none of them being recycling-oriented, promoting complex rather than linear relationships [86]. However, the author points out that examples that have the potential for expansion through bilateral exchanges are called IS precursors or clusters [86].

Most bio-based products are potentially part of the circular economy and industrial symbiosis; however, the conversion of bio-based products and waste streams into value-added products is part of the *circular economy*. In the context of the circular bioeconomy (CBE), biological resources are sustainably managed and recovered or reused when possible. Brandão et al. present the following three complementary perspectives seen as interfaces between EC and CBE: The use of biomass as a resource (by-products or waste); the sequential recycling of a material into another type of product after use; the effort to create a sustainable environment and a resource-efficient society [87].

Despite these opportunities, the way companies think and operate still needs to change considerably to address systemic challenges related to environmental conditions [88], especially considering the corporate world's position of influence in the global economy [89]. Governments also show difficulties in acting in a more incisive manner against the corporate world under the claim of a threat to continued economic growth [90]. Public funding is often relatively shortsighted, and the lack of flexibility and security can increase project uncertainty and volatility [91].

Many activities in the business sector need research for an understanding of how companies can achieve and support sustainable development in the context of their business [92]. Sachs et al. highlight six transformations needed to achieve SD, among which number three brings the decarburization of energy sources into the circuit of circularity in the management of industries, water, and waste management with the circular economy approach [93]. Section 2.2 will address the context of craft breweries and the main solid waste generated—brewers' spent grain.

### 2.2. Craft Breweries and Brewing Spent Grain Waste

The craft beer industry offers a differentiated product produced on a small scale that follows the precepts of the German beverage quality law. Data from the Ministry of Agriculture, Livestock and Supply-MALS indicates that in the year 2018, there were 889 registered breweries in Brazil [94]. In addition, by July 2020, there were 1314 registered breweries, a growth of 47.80% in less than 2 years.

Despite the effort to reduce waste in the beverage industry, a large amount remains. The processes for beer production include the following four main steps: wort preparation, fermentation, maturation, and filtration and/or stabilization [95].

The wort preparation process removes most of the BSG. Studies show that the average discard of this waste in the beer brewing process is from 14% to 20% [96,97]. This residue is now (mostly) consumed as animal feed [96,98,99], without adding value, most of it being donated to rural producers [33,100].

However, the current trend towards minimizing waste and pollution from industrial activities requires the redefinition of by-products as potential raw materials for other processes [101].

The BSG residue can be highly harmful to the environment. If discarded in rivers, it can decrease the concentration of oxygen in the site and kill important microorganisms [102].

In addition, a large number of suspended solids reduces the amount of light that can affect photosynthetic organisms [103].

However, most food by-products generated by agro-industries are sources of fiber and great importance from a nutritional point of view [104,105] and are also seen as important low-cost alternatives for food enrichment and nutritional components of the human diet [9]. BSG is a lignocellulosic material rich in fibers, proteins, and minerals [32], a source of bioactive compounds with strong antioxidants [106]. The dry material is comprised of 3.9% ash, 19.2% crude protein, 6.1% soluble lignin, 11.7% insoluble lignin, 17.9% cellulose, and 35.7% hemicellulose [30]—a source of fiber and low in carbohydrates [29–31]. Section 2.3 highlights studies on spent grain flour.

### 2.3. Brewers' Spent Grain Flour and Open Innovation

Silva et al. [107] evaluated the flour resulting from drying and milling the BSG and concluded that it presents microbiological characteristics within the standards of the Brazilian legislation, highlighting the physicochemical composition of low lipid and high protein content. Costa et al. [108] concluded that flour made from BSG shows high levels of protein, fiber, and bioactive compounds, evidencing the potential of the product as a food ingredient. It can also be used for low-gluten or gluten-free foods due to its low gluten content. Silva et al. [107] and Costa et al. [108] employed samples of Brazilian breweries. Nagy and Diósi [109] conclude that after the conversion process, BSG residue can produce a positive nutritional effect if used in the baking industry.

BSG flour can be offered to consumers directly or through the baking industry, local bakeries, and bread and cookie manufacturers. It can be used totally or partially in the food composition. Bread is a widely and universally consumed food in Brazil, accounting for up to 6% of the total calories in the Brazilian diet [110]. Nowadays, Brazilian consumers increasingly express interest in new products, variety, and innovation, especially those consistent with a healthy lifestyle [9].

The Brazilian Association of Pasta Industries–ABIMA underlines the rapid market growth of the whole grain line due to consumer demand for healthier foods [8]. Therefore, the Brazilian food industry faces the challenge of developing a variety of more nutritious products, and the literature is vast in studies on the nutritional assessment of inputs that can partially or fully replace traditional flours [111–114].

Another influential factor regards the consumption of wheat flour, the main raw material for bread, cookies, and pasta manufacturers. In Brazil, since domestic production is low, manufacturers depend on imports. Currency fluctuations, especially USD and EUR, influence input prices [115]. Data from the Brazilian Association of the Wheat Industry, ABITRIGO, reveals that in 2019, wheat flour importation accounted for 369,453 tons. [116]. The present study does not consider the 2020 report due to this being the epicenter of the COVID pandemic in Brazil.

The interconnection of these two sectors—the craft brewery and the bakery—offers an opportunity for collaboration and the generation of a new sustainable product. The analysis and detailing of the techniques and technologies used in this opportunity is a way to develop open innovation in companies. Open innovation is a paradigm that assumes that companies can and should use external ideas as well as internal ideas, and internal and external paths to market as companies seek to advance their technology [117]. The open innovation model proposed by Chesbrough emphasizes the relevance of information flows in and out across organizational boundaries; these flows are purposefully triggered to access external sources of knowledge and commerce.

While input flows enable the integration of knowledge, talent, funds, and technology into the organization, output flows enable the organization to share its knowledge, new products and services, its latest ventures, as well as intellectual property [118]. The open innovation model aims to address the traditional "black box" challenge of innovation [117]. The amount and intensity of the use of external sources define the degree of openness

(customers, suppliers, competitors, universities, research centers, etc.) throughout the innovation process [119,120].

An important idea embedded in this concept concerns the intentional management of inbound and outbound knowledge flows, which serve to accelerate internal innovation and expand external markets [45]. In summary, an OI project requires (1) a dedicated design effort before it is initiated and (2) dedicated oversight to ensure that it continues to make acceptable progress toward a high-quality outcome. Open innovation (OI) has attracted significant attention over the years, and there is considerable evidence documenting the benefits of technology companies' opening up the R&D process to external stakeholders [121].

Many governments and organizations recognize the potential of open innovation (OI) models to engage a large number of people beyond the boundaries of their organization [122]. Oliveira et al. report in their study that among the surveyed companies, the majority that perform incremental innovation perceive many constraints and that the depth of the search for external knowledge significantly influences innovation performance [123]. Open innovation practices could mitigate existing barriers; therefore, permeate the knowledge filter and theorize on the importance of institutional factors for open innovation theory in emerging economies [123].

An example of the potential of open innovation is GPS. GPS started as a project of the United States Department of Defense in 1978 and, after two decades, was made available for open global use [124]. GPS-based positioning, navigation, and timing have become the backbone of various products and services in a wide range of industries that include financial, telecommunications, transportation, agriculture, etc. [125].

In addition, the ability to identify and collaborate with external sources of knowledge and eco-innovative characteristics of SMEs is essential for policymakers and business professionals [126]. Relationship intensity or relational increase between firms can stimulate open innovation activities [127]. However, the lack of relational mechanisms makes it more difficult for collaborative partners to share their knowledge assets [128].

## 3. Materials and Methods

The research method is a quantitative simulation method, namely, the Monte Carlo method. This scientific research article uses primary data collected through bibliographic and documentary research. The survey follows the following steps described below:

The first is the location of the plant. The choice of the location of the new plant for this study relies on the distance and the number of possible suppliers of BSG, according to their characteristics, quantity, and location of the breweries. The present study used data from the Ministry of Agriculture Livestock and Supply-MALS-indicative of 2019.

After simulation of raw material supply. The supply of inputs relies on the information provided by companies to MALS 2019–2020, and seasonality estimated based on data from AMBEV Cervejas Brazil and confirmed with craft beer producers.

The third step is equipment analysis based on the supply of BSG. The humidity of BSG is based on the literature and on the instruction material of specific equipment for drying this craft beer industrial residue. The identification of the equipment is based on the cost, the capacity of the equipment for the simulated waste supply, and the possibility of using renewable energy.

After the product price simulation. The research considered a special flour, the BSG flour. The price simulation is based on substitute products with high nutritional value and rich in fiber. Four was based on Operationalization as follows: costs and operational expenses simulated based on the costs of a small transformation plant installed in Porto Alegre-RS. In addition, five-step in financial and economic analysis. Net Present Value, Internal Rate of Return, discounted payback, and Monte Carlo simulation calculations.

All values referring to price, revenues, costs, and expenses were collected in Brazilian currency (BRL) and transformed into U.S. dollars using the exchange rate valid on 10 December 2021 (BRL 5.57 per USD 1).

## 4. Analysis

### 4.1. Biorefinery Location and Raw Material Supply

Due to the characteristic of the BSG residue (namely, the time-lapse to the use before losing quality) and to ensure logistics efficiency, the distance among suppliers is an influent factor on the plant's location. MALS's brewery registration data showed that Porto Alegre (RS) and São Paulo (SP) stand out for having, respectively, 37 and 38 registered companies. Porto Alegre has a territorial extension of 496.8 km$^2$, while São Paulo has an extension of 1521 km$^2$. Therefore, the authors selected Porto Alegre as the plant's location. The city is the capital of the state of Rio Grande do Sul, in the southern region of Brazil, with an estimated population of 1.4 million in 2020. There were 14 breweries, out of the 37, located in a neighborhood less than 10 km apart from each other, justifying the choice. Based on the data provided in the breweries' annual accessory declaration to the Ministry of Agriculture, Livestock and Supply (regarding 2019), the amount of waste volume was estimated. The investigation does not consider the 2020 report because several breweries stopped their activities during the most critical period of the COVID pandemic in Brazil.

The research considers a waste generation range of 14 kg to 20 kg per 100 L of beer [96,97]. This way of simulating the by-products was necessary since the producers did not have control over the waste generation of their production.

As estimated in Appendix A, the joint annual amount of waste generated in 2019 by the 14 breweries is 170,959 kg to 244,227 kg. Craft breweries generally produce on supply without much storage volume. The Ambev Cervejas Brazil (a benchmark company in the industry) report for 2017–2019 provided the volume of beer sales in Brazil, required to verify the production seasonality and consequent waste availability. The producers confirmed the estimate made according to their experience, since most of them did not have reliable control systems. Appendix B shows the production volume and the calculation of the percentage per quarter.

Beer production and consumption in Brazil have seasonality marked by quarters. Higher production is marked by the months from October to December, with the beginning of summer in the Global South and New Year festivities. The same is true in the first quarter (Jan–Mar), due to the vacation period, summer, and carnival festivities. Thus, the percentages of annual sales are 25%, 22%, 23%, and 30%, respectively, for the first, second, third, and fourth quarters. These percentages support the minimum and maximum quarterly calculation of the waste supply, which determines the flour production process. Table 1 presents the estimated minimum and maximum amounts of BSG raw material per quarter.

**Table 1.** Supply of spent grain by quarter.

| Quarter | % | Minimum (14%) | Maximum (20%) |
|---|---|---|---|
| 1º | 25% | 42,740 | 61,057 |
| 2º | 22% | 37,611 | 53,730 |
| 3º | 23% | 39,321 | 56,172 |
| 4º | 30% | 51,288 | 73,268 |
| Total | 100% | 170,959 | 244,277 |

The authors researched the necessary equipment according to a pull and intermittent production system based on the minimum and maximum available raw materials. The supply is uncertain from the aspect of the amount of waste generated by each brewery. For example, a brewery can generate from 15% to 17% of waste. For this reason, for the revenue-forming variables (price and supply), the investigation used the Monte Carlo simulation. Another factor that can increase the supply of waste is the amount of beer produced, which follows a growing trend of new enterprises and consumers of craft beer.

*4.2. Production System*

The production process begins by collecting the spent grain in barrels, transferring it to pressing machines to reduce the percentage of liquid, and then entering the drying process. Thus, it reaches the necessary humidity to be ground and is then transformed into BSG flour. Next, in Section 4.2.1, we verify the equipment and its production capacity.

### 4.2.1. Equipment

For the operationalization of the biorefinery, the authors conducted a survey with three manufacturers of specific equipment for drying agricultural products, specifying the characteristics of the input. One of the consulted companies projected specific equipment for drying spent grain, but the high cost of the equipment compared to the financial capacity of the breweries makes the sale of the product unfeasible. The research uses four pieces of equipment for the operationalization of the biorefinery. Table 2 shows the equipment and its respective function.

**Table 2.** Equipment and your function.

| Equipment | Function |
|---|---|
| Mechanical press (Hydrus) | To reduce the humidity of the residues and facilitate product drying; |
| Hybrid dryer (Hybrid Dryer) | To perform the final drying of the product; |
| Mill (Mill MCS 280) | To grind the product and transform it into flour; |
| Packing machine (Automatic Packing Machine) | Packing materials that can be used in this equipment include paper, aluminum, polyethylene, others. |

The four pieces of equipment together, considering the costs of importing the hybrid dryer to Brazil, come to USD 87,753. The main advantage of hybrid dryers over solar dryers is the possibility to control the drying process since the auxiliary energy system maintains uniform drying conditions. In addition, regarding artificial dryers, they present economic advantages since they operate with a renewable source as the main energy source.

### 4.2.2. Operational Capacity of the Equipment

The mechanical press is capable of reducing product moisture by 30%. After this reduction, the pre-treated spent grain goes to the hybrid dryer with a capacity of 850 kg of wet material (estimated input moisture of 55–60%).

To calculate the weight of the residue entering the dryer, the formula below was used to calculate the subtraction from the loss of mass due to drying (wet mass loss), since the loss of moisture is not linear to the loss of weight [129].

$$HB = \left[ \frac{Ih - Fh}{100 - Fh} \right] * 100, \tag{1}$$

where:
HB = % Humidity breaker
Ih = Initial humidity
Fh = Final humidity

Table 3 shows the calculation of the dryer activation within the quarter and the estimate for each month within that quarter due to the minimum and maximum supply of the pretreated spent grain residue. To calculate the weight loss due to the loss of humidity, the initial humidity was set at 88% and the final humidity at 58%.

The dryer proves to be sufficient for the supply of inputs, considering its 2-day cycle. Establishing a base month of 30 days, the maximum limit of the dryer would be 15 days. At the maximum supply in the fourth quarter, the equipment would be activated 25 times within the availability of 90 days. Thinking in monthly terms, in October, for example, it would be driven 9 (8.2) times, which reflects the maximum consumption of 20 days, while

counting the other 10 days as idle or even in the event of a pause for rest/maintenance of the equipment.

**Table 3.** Dryer operation cycle.

| Quarter | Minimum | Maximum | Weight (in kg) with the Loss of Humidity in the Press | | Dryer Drive per Quarter | | Dryer Drive per Month | |
|---------|---------|---------|---------------|---------------|---------|---------|---------|---------|
| | | | Minimum Supply | Maximum Supply | Minimum | Maximum | Minimum | Maximum |
| 1° | 42,740 | 61,057 | 12,211 | 17,445 | 14 | 21 | 4.8 | 6.8 |
| 2° | 37,611 | 53,730 | 10,746 | 15,351 | 13 | 18 | 4.2 | 6.0 |
| 3° | 39,321 | 56,172 | 11,235 | 16,049 | 13 | 19 | 4.4 | 6.3 |
| 4° | 51,288 | 73,268 | 14,654 | 20,934 | 17 | 25 | 5.7 | 8.2 |

Due to the high moisture concentration, the life of the residue is from 7 to 10 days [57]. Thus, the original spent grain can be stored to complete the capacity of the equipment if the need arises. However, the authors state that the durability depends on the chemical composition of the BSG, which can change depending on the barley used, harvest time, malting, and mixing time. For storage, the authors recommend a moisture content of 10%. However, for milling, the time estimated by the drying equipment manufacturer will be followed (6–8%).

When leaving the dryer, the product has an estimated humidity of anywhere from 6 to 8%. We recalculated the weight. Table 4 shows the weight calculation for the milling cycle. For the calculation, the initial moisture was set at 58% and the final at 6%.

**Table 4.** Calculating weight for grinding.

| Quarter | Minimum | Maximum | Weight Loss Calculation (in kg) | | | |
|---------|---------|---------|----------------|----------------|---------|---------|
| | | | Minimum Supply | Maximum Supply | per Month | |
| 1° | 12,211 | 17,445 | 5456 | 7795 | 1819 | 2598 |
| 2° | 10,746 | 15,351 | 4801 | 6859 | 1600 | 2286 |
| 3° | 11,235 | 16,049 | 5020 | 7171 | 1673 | 2390 |
| 4° | 14,654 | 20,934 | 6547 | 9353 | 2182 | 3118 |

The third stage is milling, but after drying, the product can already be stored for 6 months (dry grain and flour). The mill has an operational production process of screening 0.3 mm–30 kg/h (approximately). The operational capacity of the mill is (30 kg × 220 h) 6600 kg/month, within the plant's needs, where the estimated maximum processing weight is 3118 kg. The workload of 220 h per month is under the Brazilian labor legislation for monthly working hours per employee.

*4.3. Flour Production-Revenue Estimation-Operational Expenses*

The definition of the quantity of flour production considers a 5% loss estimate. This breakage is attributed to the handling and packaging processes of the product.

As for the price per kg of flour, for initial comparison purposes, only one company producing spent grain flour was found. The company located in the United States performs the sale through its website, and the Brazilian consumer can purchase 4.5 kg for USD 110 or 2.27 kg for USD 60 [130]. On average, USD 25 per kg, well above the maximum value proposed of USD 3.60 per kg. This company performs the production of flour in an artisanal way, in small quantities. Due to the difference in quantity proposed here and the location of this company, we opted to set a conservative price based on substitute products. Below is a description of some of the items consulted on the website of a Brazilian company. Table 5 shows the data collected.

**Table 5.** Description of some of the items consulted.

| Special Flour | USD (per kg) |
|---|---|
| Passion fruit flour | 3.44 |
| Chickpea flour | 2.69 |
| Green banana flour | 3.59 |

The authors considered specialty flours rich in fiber and nutrients as substitute products. Their prices vary between USD 2.70 and USD 3.60 p/kg.

In addition, the operation of the bio-refinery, according to the workflow, requires three production employees and two administrative employees (operating expenses). In addition, it will have expenses for energy, water, packaging materials, transport maintenance, fuel, and rent. To illustrate, the total monthly expenses of a small fruit drying industry in Porto Alegre-RS are 3536 USD, and Appendix C presents the details of the values.

As for the variable expenses, such as taxes and commissions, a percentage of 20% over gross revenue was considered (Simples Nacional industry table II-band III-taxes 10% over gross revenue + 10% commissions on sales to breweries-payment of raw material).

A commission on revenues is proposed as payment for the raw material—the spent grain by-product. This is expected to motivate the brewers to dispose of their waste at the plant. However, this study does not focus on developing a business model; as such, the aforementioned factor only serves as an example of the viability of the plant.

Table 6 presents the calculations regarding the production drop and the minimum and maximum revenue per quarter, as well as the simulation of payment for the raw material spent grain. However, it is necessary to recognize the economic benefits beyond monetary rewards; indirect financial approaches need to be considered. The image of companies in the sector, indirect marketing, and social responsibility are some of the aspects that are easily evidenced. The visualization and knowledge of these data is an incentive factor to share the internal knowledge of companies, meeting open innovation and industrial symbiosis, which both potentiate a circular economy model and, consequently, the sustainable development so desired.

**Table 6.** Production/Simulated revenue per scenario and simulated return per brewery with 10% commission.

| Quarter | Production kg | | Production kg with Loss | | Revenue per [1] Scenario (Minimum Price USD 2.70–Maximum Price USD 3.60) | | | |
|---|---|---|---|---|---|---|---|---|
| | Minimum | Maximum | Minimum | Maximum | Scenario 1 | Scenario 2 | Scenario 3 | Scenario 4 |
| 1 | 5456 | 7795 | 5183 | 7405 | USD 13,994 | USD 18,659 | USD 19,994 | USD 26,658 |
| 2 | 4801 | 6859 | 4561 | 6516 | USD 12,314 | USD 16,419 | USD 17,593 | USD 23,457 |
| 3 | 5020 | 7171 | 4769 | 6812 | USD 12,876 | USD 17,168 | USD 18,393 | USD 24,524 |
| 4 | 6547 | 9353 | 6220 | 8885 | USD 16,793 | USD 22,390 | USD 23,990 | USD 31,987 |
| Total | 21,824 | 31,178 | 20,733 | 29,619 | USD 55,978 | USD 74,638 | USD 79,971 | USD 106,628 |

| | Scenario 1 | Scenario 2 | Scenario 3 | Scenario 4 |
|---|---|---|---|---|
| Revenue | USD 5597 | USD 7463 | USD 7997 | USD 10,662 |
| Breweries [2] | 14 | 14 | 14 | 14 |
| Average revenue | USD 399 | USD 533 | USD 571 | USD 761 |

[1] Scenario 1: Minimum production and minimum price; Scenario 2: Minimum production and maximum price; Scenario 3: Maximum production and minimum price; Scenario 4: Maximum production and maximum price. [2] Number of breweries in this study.

### 4.4. Financial and Economic Feasibility Analysis

The net present value and rate of return analysis used the following parameters:

- Minimum rate of attractiveness (MRA) = 5% p.a.;
- Initial investment: working capital + equipment = USD 3590 + USD 87,753 = USD 91,343;

- Monthly expenses = USD 3536;
- Analysis time: 5 years (no residual value);
- Calculated revenues in four scenarios: (i) Minimum production and minimum price; (ii) minimum production and maximum price; (iii) maximum production and minimum price; (iv) maximum production and maximum price.

The first scenario presents a negative NPV in the five-year analysis, and the capital return occurs after ten years of plant operation. In the second scenario, maintaining minimum production and changing the price to the maximum proposed, the NPV is positive at USD 19,220, and the internal rate of return in the five years is 12.67%, surpassing the proposed TMA, and the capital return occurs in four years.

The second and third scenarios simulated maximum production. The NPV is positive by USD 36,800 in the lowest price scenario and reaches USD 229,732 in the highest price scenario. The IRR is 19% and 79%, respectively, and the return on investment is 3.2 years for the lowest price and 1.2 years for the highest price.

Except for the first scenario with minimum production and minimum price, the other results are in line with the research of Swart et al. [131], who investigated the valorization of BSG in a small-scale biorefinery located in an annex of a brewery. The authors verified the feasibility of BSG conversion in the following three different scenarios: sugar substitute xylitol; prebiotic xylo-oligosaccharide-XOS; coproduction of xylitol and XOS. In the study, the three scenarios obtained exceeded the proposed MTE of 9.7%.

After the simulation of the static scenarios with the mentioned data, the simulation was performed by the Monte Carlo method (random simulation), which included 10,000 runs using Excel 2013 and normal distribution.

The quantitative variables, price and supply, are ordinal in scale. They vary randomly from a minimum to a maximum level. The parameters used in the first simulation were the following:

- Supply in kg: 20,733–29,619;
- Price USD: 2.70–3.60;
- Taxes and commissions: 20% on revenue;
- Expenses: USD 3536 monthly–USD 42,441 annually;
- Profit is obtained with Equation (2):

$$\text{Profit} = ((\text{Supply} \times \text{Price}) - ((\text{Supply} \times \text{Price}) \times 20\%)) - \text{Expenses} \qquad (2)$$

In this simulation, in the 10,000 runs there was no loss the minimum profit was USD 2513, the maximum profit was USD 42,542, and the average profit was USD 20,953. In this simulation, the equilibrium supply point considering the minimum price is 19,649 kg/year. From this simulation, the researchers found the percentage of return on investment in years based on the frequency of profit in blocks. Appendix D presents the profit range and frequency.

Table 7 shows a summary of the results regarding the financial and economic analysis (net present value, internal rate of return, and payback) of the static scenarios and the Monte Carlo simulation results.

The authors used a discount rate of 5% p.a. to estimate the return time of the investment. Therefore, it is evident that in 7% of the interactions, the return on investment occurs within 2 years, 31.6% within 3 years, 64% within 4 years, and 84% within 5 years.

To analyze the sensitivity of supply and price, the probability of loss was verified by keeping one variable random and fixing the other. The reduction of the fixed variable was also 10%. The supply proved to be more sensitive. With a 10% reduction in supply, the probability of loss was 12%, while with a price reduction, and the percentage was 10%. With a 20% and 30% reduction in supply, the probability of loss was 58% and 97%, respectively. In the reduction of price in the same 20% and 30%, the probability of loss was 40% and 87%, respectively.

**Table 7.** Summary of the financial and economic analysis and Monte Carlo simulation.

| | Financial and Economic Analysis | | | |
|---|---|---|---|---|
| **Scenarios** | **Scenario 1** | **Scenario 2** | **Scenario 3** | **Scenario 4** |
| NPV | −USD 42,335 | USD 19,220 | USD 36,800 | USD 229,732 |
| IRR | −15.23% | 12.67% | 19.22% | 79.59% |
| Payback (year) | 10 | 4 | 3.2 | 1.2 |
| | **Monte Carlo Simulation** | | | |
| | **Initial Data** | | | |
| **Average Profit** | **Standard Deviation** | **Probability of Loss** | **Minimum Profit** | **Maximum Profit** |
| USD 20,953 | USD 8346 | 0.001 | USD 2513 | USD 42,542 |
| | **Time Percentage of Return on Investment-Discount Rate 5% p.a.** | | | |
| Within 2 years | | 7% | Within 4 years | 64% |
| Within 3 years | | 31% | Within 5 years | 84% |
| | **Supply × Price Sensitivity** | | | |
| **Supply kg *** | | **Probability of Loss** | **Price kg **** | **Probability of Loss** |
| 20,733 | | 0% | 2.70 | 0% |
| 18,659 | | 12% | 2.43 | 10% |
| 16,586 | | 58% | 2.16 | 40% |
| 14,513 | | 97% | 1.89 | 87% |

\* Fixed price-random price (USD 2.70–USD 3.60). \*\* Fixed price-random offer (20,733–29,619 kg).

## 5. Discussion: The Conversion of Brewers' Spent Grain into a Special Flour, and Open Innovation

The implication of the study is the possibility of developing a new, innovative product relying on the concept of open innovation, aiming at supporting the decision-making process in artisanal SMEs (small and medium enterprises) that employ raw materials retrieved from other industries [10]. Open innovation concerns embrace the development and commercialization of innovative products in innovative formats, such as licensing agreements or startup projects. Open innovation-based products rely on the joint development and combination of internal and external ideas [132], such as those presented in this article, to result in a new product that overcomes the usual difficulties faced by managers in decision-making processes related to waste management.

To institutionalize open innovation research, highlights the importance of the value of using non-cash rewards and informal controls to ensure that OI creates value for stakeholders [121]. In the non-pecuniary mode of OI input occurs the acquisition of external knowledge without there being necessarily a compensation of outside ideas and financial contributions [133].

There is a need to better understand the effective use of a collaborative and open innovation approach in research and management focused on environmental sustainability [134]. Showing projects that are economically viable and that offer environmental and social benefits could encourage new designs in the area of open innovation. Since studies reveal that even SMEs prefer to be closed to protect know-how in the experimentation phase [51] or the reduced openness of SMEs through a lack of infrastructure and financial resources [135].

In this model, the biorefinery is open data, presenting its techniques and processes. Its output stream is also open, and at the same time, it launches reflection for the necessary me-improvement for higher profitability of the business. The output streams allow the organization to share its knowledge, new products and services, latest ventures, as well as intellectual property [118].

Research on open innovation clearly points to the importance of factors that relate to knowledge or relational mechanisms [128,136,137]. In this sense, industrial symbioses strengthen open innovation because relationships between firms are one of the basic re-

quirements of IS, and at the same time, knowledge sharing between firms enables open innovation. Open innovation can effectively deal with resources and the environmental externalities and then relatively balance the economic value and green value of organizations [43]. Biorefinery becomes the bridge between economic value and the environment, joining IS and open innovation.

## 6. Conclusions

This study aimed to analyze the financial and economic feasibility of implementing a biorefinery for flour production based on the spent grain waste generated in the craft beer brewing process by industries in Porto Alegre-RS.

Individually, the cost of transforming the spent grain waste into flour is unfeasible due to the amount of investment required for equipment. In addition, brewers have their attention focused on new flavors and aromas for their products. This makes the by-product not valued, being most of the time donated to rural producers for complementing animal feed or simply discarded as urban garbage (due to the small individual proportion produced). In the latter case, it causes a burden on the public coffers responsible for the collection and disposal of waste.

Aiming at finding an economically viable solution, this study carried out simulations to analyze the implementation of a plant for the collective transformation of these by-products and thus add value to the by-product. The present scientific research article presents the waste transformation process based on the processing through three pieces of equipment with different production capacities. The research analyzed the cycle of each piece of equipment in an intermittent production system. It was found that the equipment could indeed process the simulated supply of inputs from the industry. The authors chose the equipment considering the factors of savings, cost, and use of renewable energy.

The results demonstrate the feasibility of three static scenarios presented with positive NPV and IRR above the minimum rate of attractiveness in the cycle with projections for up to five years. The Monte Carlo simulation demonstrated that in 64% of the interactions performed, the return of capital occurs within four years. However, the supply sensitivity is a little higher than the price sensitivity.

The data presented in this research allows the reflection of new products, businesses, and processes. Open innovation finds support from scientific research and can be a point of support for small businesses, stimulating the reflection of possibilities, as in the case presented in the Porto Alegre craft breweries cluster. The proposed plant would also offer new possibilities.

The dryer could be used in the idle period to benefit other products, such as drying fruit, since the plant is located near a fruit distribution center. Another possibility is the reuse of the liquid residue resulting from the process of pressing and drying the spent grain.

These results may encourage new investors who seek a business segment that values residues and who seek to capitalize on a promising market of customers seeking more products that are nutritious. Allied to this is the evident pressure felt by governments to seek a paradigm shift in the ways of production and consumption, as governments are increasingly expected to implement new policies to support socio-environmental entrepreneurship. Furthermore, transforming waste into raw material for a new use could benefit breweries in the promotion of their products since the flour would have the nutritional value provided by inputs of the craft beer brewing process.

The main limitation relates to the research method, specifically the simulation. As for indications for further research, it is necessary to consider the business model and financial gains for artisanal breweries. In addition, future research should target the control of the quantities of different types of waste used in each brewery. Lastly, future research should also target how to increase revenue with liquid waste. One implication of this study was to initiate a discussion on how to operationalize the use of waste from small and medium-sized craft breweries, which generate a lower flow of waste when compared to traditional breweries.

Future research suggests testing new forms of the drying process, such as the use of passive solar energy in sunnier regions of Brazil. In this way, the drying of BSG would be similar to the drying of coffee or cocoa beans, and this could enhance the economic return. Still, brewery residues could be used for other products, as a source of raw material for biogas generation and secondary fuel. The aforementioned innovative product could also help analyze other available waste, such as biomass, mainly employed in the chemical industry [138], as well as metallic swarf and scrap, mainly generated in multiple companies in the mechanical industry [139] and recycled in large units in the steel-making industry [27].

**Author Contributions:** Investigation, I.C. and M.S.d.L.; Methodology, D.R.R.; Supervision, M.E.S.M. and M.A.S. All authors have read and agreed to the published version of the manuscript.

**Funding:** This research was funded by Federal University of Santa Maria grant number 23081.009327/ 2021-22. And The APC was funded by UNISINOS.

**Institutional Review Board Statement:** Not applicable.

**Informed Consent Statement:** Not applicable.

**Conflicts of Interest:** The authors declare no conflict of interest.

## Appendix A

**Table A1.** Simulation Brewer's Spent Grain (BSG).

| Breweries | Production (l) 2019 | Waste-kg (Minimum 14%) | Waste-kg (Maximum 20%) |
|---|---|---|---|
| 1 | 35,180 | 4925 | 7036 |
| 2 | 44,628 | 6248 | 8926 |
| 3 | 45,630 | 6388 | 9126 |
| 4 | 53,050 | 7427 | 10,610 |
| 5 | 53,750 | 7525 | 10,750 |
| 6 | 55,400 | 7756 | 11,080 |
| 7 | 66,000 | 9240 | 13,200 |
| 8 | 69,815 | 9774 | 13,963 |
| 9 | 77,638 | 10,869 | 15,528 |
| 10 | 93,835 | 13,137 | 18,767 |
| 11 | 108,550 | 15,197 | 21,710 |
| 12 | 116,000 | 16,240 | 23,200 |
| 13 | 161,160 | 22,562 | 32,232 |
| 14 | 240,500 | 33,670 | 48,100 |
| Total | 1,221,136 | 170,959 | 244,227 |

## Appendix B

**Table A2.** AMBEV Quarterly Production Volume-Brazil Beers.

| Quarter | Volume | % | Quarter | Volume | % | Quarter | Volume | % | Average% |
|---|---|---|---|---|---|---|---|---|---|
| 1T2017 | 20,549 | 26 | 1T2018 | 18,879 | 24 | 1T2019 | 21,003 | 26 | 25 |
| 2T2017 | 17,430 | 22 | 2T2018 | 17,729 | 23 | 2T2019 | 18,245 | 23 | 22 |
| 3T2017 | 18,486 | 23 | 3T2018 | 17,912 | 23 | 3T2019 | 17,417 | 22 | 23 |
| 4T2017 | 23,768 | 30 | 4T2018 | 23,264 | 30 | 4T2019 | 23,598 | 29 | 30 |
| Total | 80,234 | 100 | Total | 77,784 | 100 | Total | 80,264 | 100 | 100 |

## Appendix C

**Table A3.** Fixed operating expenses.

| Costs/Expenses | Monthly Amount |
|---|---|
| Manufacturing Employees | USD 807 |
| Administrative Staff | USD 646 |
| Energy/Water | USD 987 |
| Packaging Material | USD 179 |
| Transport Maintenance | USD 89 |
| Fuel | USD 179 |
| Rent | USD 646 |
| Total | USD 3536 |

## Appendix D

**Table A4.** Profit range and frequency in Monte Carlo simulation.

| Bloco | Frequência |
|---|---|
| 0–2500 | 0 |
| 2501–5000 | 117 |
| 5001–7500 | 341 |
| 7501–10,000 | 505 |
| 10,001–12,500 | 675 |
| 12,501–15,000 | 913 |
| 15,001–17,500 | 1002 |
| 17,501–20,000 | 1160 |
| 20,001–22,500 | 1120 |
| 22,501–25,000 | 1011 |
| 25,001–27,500 | 839 |
| 27,501–30,000 | 698 |
| 30,001–32,500 | 574 |
| 32,501–35,000 | 451 |
| 35,001–37,500 | 322 |
| 37,501–40,000 | 198 |
| 40,001–42,500 | 73 |
| 42,501–45,000 | 1 |
| Total | 10,000 |

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
