# Peer review of "Economic and Financial Feasibility of a Biorefinery for Conversion of Brewers’ Spent Grain into a Special Flour"

_2199-8531, doi:10.3390/joitmc8020079_

Round 1

Reviewer 1 Report

Overall, the paper is well written.  I have marked locations where words can be removed, in order to make the English smoother.

On page 4 - 2 sentences are highlighted.  They say that BSG are rich in sugars; yet the next sentence indicates that high levels of cellulose/hemicellulose and lignin.  They two sentences don't match.  The sugars are extracted from the grain in the brewing process, so BSG should not be rich in sugars.

The cost and economic figures should not show the number behind the decimal points.  Indeed, the cost and rates of return should consider the degree of accuracy of the results - is 4 figures of accuracy approriate?  More likely, 2 figures of accuracy is reasonable (accuracy to 1%), for such a study - and it does not detract from the conclusions.

It would seem that the major variable cost is that of energy to evaporate water from the BSG.  If it were possible to use passive solar to heat and reduce the energy consumption - then a business may make sense.  In this way - drying of BSG would be similar to drying of coffee or cocoa beans. This may be worth some reflection and addition to the conclusions.

Author Response

Overall, the paper is well written.  I have marked locations where words can be removed, in order to make the English smoother.

Answer: Thank you, we removed the words marked.

On page 4 - 2 sentences are highlighted.  They say that BSG are rich in sugars; yet the next sentence indicates that high levels of cellulose/hemicellulose and lignin.  They two sentences don't match.  The sugars are extracted from the grain in the brewing process, so BSG should not be rich in sugars.

Answer: Text adjusted. BSG are rich in fibers not in sugars.

The cost and economic figures should not show the number behind the decimal points.  Indeed, the cost and rates of return should consider the degree of accuracy of the results - is 4 figures of accuracy approriate?  More likely, 2 figures of accuracy is reasonable (accuracy to 1%), for such a study - and it does not detract from the conclusions.

Answer: We adjusted number of decimal points, costs and percentage.

It would seem that the major variable cost is that of energy to evaporate water from the BSG.  If it were possible to use passive solar to heat and reduce the energy consumption - then a business may make sense.  In this way - drying of BSG would be similar to drying of coffee or cocoa beans. This may be worth some reflection and addition to the conclusions

Answer: Yes, we included a sentence in the conclusions.

Reviewer 2 Report

The article and the problem is quite interesting. The research are well prepared.

Nevertheless, I have 1 main objection and a question? How the content connects to the scope of the journal. My guess is that there is a link, but there is no direct reference to open innovation or innovation in the text. So at this stage it needs to be completed. In my opinion, this is a necessary condition for further processing and reviewing the article. 

Author Response

Nevertheless, I have 1 main objection and a question? How the content connects to the scope of the journal. My guess is that there is a link, but there is no direct reference to open innovation or innovation in the text. So at this stage it needs to be completed. In my opinion, this is a necessary condition for further processing and reviewing the article. 

Resposta: We consider the study as a way for SMEs to pursue open innovation. We seek to adapt the text and justify the connection with the area of open innovation.

Reviewer 3 Report

Dear Author(s) I find the paper completed and much interestring.

The Introduction is strong and linked well with previous studies the methodology is clear and the quality of presentation of the results is very High.

I have only a minor concern regarding the concept of sustainable developement please, improve this aspects.

Author Response

I have only a minor concern regarding the concept of sustainable developement please, improve this aspects.

Answer: We have included the UN definition to improve the definition

Round 2

Reviewer 2 Report

The author made all needed corrections. The article could be published in present form.